# CaptainGAN: Navigate Through Embedding Space For Better Text Generation

## Abstract

Score-function-based text generation approaches such as REINFORCE, in general, suffer from high computational complexity and training instability problems. This is mainly due to the non-differentiable nature of the discrete space sampling and thus these methods have to treat the discriminator as a reward function and ignore the gradient information. In this paper, we propose a novel approach, CaptainGAN, which adopts the straight-through gradient estimator and introduces a "re-centered" gradient estimation technique to steer the generator toward better text tokens through the embedding space. Our method is stable to train and converges quickly without maximum likelihood pre-training. On multiple metrics of text quality and diversity, our method outperforms existing GAN-based methods on natural language generation.

## 1 Introduction

Generative Adversarial Networks (GAN) (Goodfellow et al., 2014) have led to many advances in image generation, image editing, style transfer, and representation learning (Karras et al., 2017; Brock et al., 2018; Karras et al., 2019). Unsurprisingly, much effort has been devoted to adopting the GAN framework for unsupervised text generation (Yu et al., 2017; Che et al., 2017; Balagopalan et al., 2018; Fedus et al., 2018; Guo et al., 2018; de Masson d'Autume et al., 2019; Zhang et al., 2017; Nie et al., 2019). However, as finding a Nash equilibrium is not as straightforward as finding a local optima, researchers have been forced to develop many ad-hoc tricks and techniques to make GAN training well-behaved. In the text generation setting, they are also faced with the additional obstacle of passing discrete tokens through a non-differentiable operation, which prohibits back-propagating the gradient signal to the generator.

To address the issue of non-differentiablity, researchers and practitioners use score function gradient estimators such as REINFORCE to train GANs for text generation, where the discriminator is cast as a reward function for the generator. However, these methods still suffer from poor sample efficiency due to the credit assignment problem. We argue that it is disadvantageous to utilize the discriminator as simply a reward function when it is known that gradient-based backpropagation is a far more efficient way to perform credit assignment.

In this paper, we propose a novel unsupervised text generation technique, called CaptainGAN, which propagates a modified gradient signal from discriminator to generator in order to improve the efficiency and accuracy of the estimator. Our contributions are as follows:

- An update procedure for the generator to incorporate gradient information from the discriminator during generator training.

- Lower memory and computational requirements than other RL-based counterparts.

- Near SOTA results without maximum likelihood pretraining.

Please see Appendix A for a detailed description of the notation used in this paper.

## 2 BACKGROUND

The Generative Adversarial Network (GAN) proposed in Goodfellow et al. (2014) is an innovative approach to the generative modeling problem. Rather than using the maximum likelihood estimation (MLE) directly to learn a probabilistic model, the GAN is a two-player minimax game in which the goal of one player, the generator $G_\theta$, is to generate samples $\hat{\mathbf{x}}$ from $p_\theta$, and the goal of the other player, the discriminator $D_\phi$, is to learn to classify whether or not a sample was generated from real data $p_{data}$ or the generator.

$$\mathcal{V}_D = \mathbb{E}_{\mathbf{x} \sim p_{data}}[\log D_\phi(\mathbf{x})] + \mathbb{E}_{\hat{\mathbf{x}} \sim p_\theta}[\log(1 - D_\phi(\hat{\mathbf{x}}))] \tag{1}$$

$$\mathcal{V}_G = \mathbb{E}_{\hat{\mathbf{x}} \sim p_\theta}[\log(D_\phi(\hat{\mathbf{x}}))] \tag{2}$$

where $\mathcal{V}_D$ and $\mathcal{V}_G$ are respectively the objective functions of the discriminator $D_\phi$ and the generator $G_\theta$. Equation 2 is the *alternative generator loss* suggested by the original work as its gradient does not vanish when $D_\phi(\hat{\mathbf{x}})$ is small.

In the standard GAN architecture, the generator's output is directly connected as the input to the discriminator in a fully differentiable manner, which means the gradients from the discriminator's loss function can be back-propagated to the parameters of the generator. However, it requires sampling a sequence of tokens from a discrete distribution in text generation, which is essentially non-differentiable. In order to avoid the intractability of the gradients, text GANs resort to some sort of estimation.

### 2.1 CONTINUOUS RELAXATION

Continuous relaxation approaches such as Gumbel-Softmax (Jang et al., 2016) approximate a stochastic categorical distribution in terms of a deterministic continuous function. While this apparently allows us to remove the non-differentiable discrete sampling altogether (Kusner & Hernández-Lobato, 2016; Nie et al., 2019), it creates several serious issues.

The continuous distribution generates the expectation of embeddings - a weighted sum with no direct correspondence to an exact word or token. This means all the discriminator has to do is spot the difference the actual word embeddings and expectation. In turn, the generator will try to compensate by producing extremely "spiky" predictions.

Furthermore, this way of generating data creates a major inconsistency in that during inference , the generator has to sample a discrete sequence from distribution whereas during training, it's only trained to generate an expectation that is feasible to discriminator.

### 2.2 SCORE-FUNCTION GRADIENT ESTIMATOR

The score-function gradient estimator (Fu, 2006; Glynn, 1990), also known as the REIN-FORCE (Williams, 1992) is a common solution for non-differentiable issue as mentioned above. Applying the REINFORCE algorithm, the gradient of the expectation of reward function $f_\phi$ can be written as

$$\frac{\partial}{\partial \theta} \mathbb{E}_{\hat{\mathbf{x}} \sim p_\theta}[f_\phi(\hat{\mathbf{x}})] = \mathbb{E}_{\hat{\mathbf{x}} \sim p_\theta}[f_\phi(\hat{\mathbf{x}}) \frac{\partial}{\partial \theta} \log p_\theta(\hat{\mathbf{x}})]. \tag{3}$$

Since it does not require $f_\phi$ to be differentiable or even continuous as a function of $x$, the gradient of $\mathbb{E}_{\hat{\mathbf{x}} \sim p_\theta}[f_\phi(\hat{\mathbf{x}})]$ can be back-propagated to the generator $G_\theta$. In the context of GANs, $\mathbb{E}_{\hat{\mathbf{x}} \sim p_\theta}[f_\phi(\hat{\mathbf{x}})]$ can be seen as the objective function of the generator and the reward function $f_\phi$ can be replaced with $\mathcal{D}_\phi$.

Although REINFORCE is an unbiased estimator, it still has a number of disadvantages such as high variance, low sample efficiency and the credit assignment problem. Therefore, much effort is devoted to reducing the variance using special methods (Gu et al., 2016; Grathwohl et al., 2017), or to providing more dense rewards such as in Yu et al. (2017); Che et al. (2017); Fedus et al. (2018); de Masson d'Autume et al. (2019), where Monte-Carlo roll-outs are used to obtain per-word rewards.

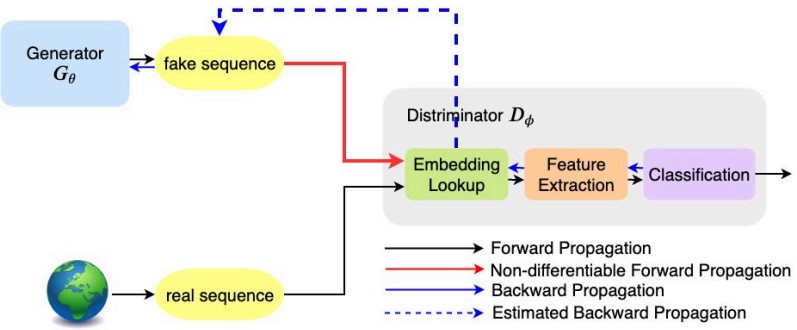

Figure 1: The non-differentiable operation between the generator and the discriminator.

### 2.3 Straight-Through Gradient Estimator

Another approach is using a straight-through gradient estimator (Bengio et al., 2013; Jang et al., 2016). The basic idea is to perform non-differentiable operation during the forward pass, but *approximate* it with a differentiable proxy during the backward pass.

Consider sampling from a categorical distribution (notation borrowed from Jang et al. (2016)): The following operation is performed during the forward pass

$$\mathbf{z} = \text{one-hot}(\hat{x}) \tag{4}$$

where a categorical sample $\hat{x}$ is encoded as a $v$-dimensional one-hot vector $\mathbf{z}$ ($v$ being the given vocabulary size). During the backward pass, the corresponding straight-through estimator of function $f$ with respect to $\theta$ is

$$\hat{g}_\theta = (\frac{\partial f}{\partial \mathbf{z}})^\top \frac{\partial \mathbf{m}_\theta}{\partial \theta} \tag{5}$$

derived by using the approximation $\frac{\partial \mathbf{z}}{\partial \theta} \approx \frac{\partial \mathbf{m}_\theta}{\partial \theta}$, where $\mathbf{m}_\theta$ is a differentiable proxy for the non-differentiable one-hot encoding vector $\mathbf{z}$ (Jang et al., 2016). Moreover, Jang et al. (2016) suggest the choice of

$$\mathbf{p}_\theta = [p_\theta(x_1), \ldots, p_\theta(x_v)]^\top \tag{6}$$

as proxy, where $\{x_1, \ldots, x_v\}$ are all possible tokens in vocabulary $V$. Compared to the former method, this approach updates all the possible tokens' probability, even if the tokens were not sampled.

## 3 Proposed Method

The non-differentiable connection between the generator $G_\theta$ and the discriminator $D_\phi$ is our primary focus in this paper (Figure 1). Starting from the straight through gradient estimator, we derive a re-centered estimator which helps the generator learn to select better tokens.

### 3.1 Gradient Estimation in the Generator

Consider a discriminator whose first layer is an embedding lookup layer. During forward propagation, the operation below is performed first:

$$\mathbf{e}_{\hat{x}} = E.lookup(\hat{x}) = E\mathbf{z} \tag{7}$$

where $E = [\mathbf{e}_1, \ldots, \mathbf{e}_v]$ and $\mathbf{z}$ is one-hot encoding of vocabulary. As described in Section 2.3, this is a non-differentiable categorical sampling operation with a straight-through estimator

of the gradient of $\mathcal{V}_G$ with respect to $\theta$, formulated as follows:

$$\hat{g}_\theta = \sum_{t=1}^{T} \left(\frac{\partial \mathcal{V}_G}{\partial \mathbf{z}_t}\right)^\top \frac{\partial \mathbf{p}_{t,\theta}}{\partial \theta} = \sum_{t=1}^{T} \left(E^\top \frac{\partial \mathcal{V}_G}{\partial \mathbf{e}}(\mathbf{e}_{\hat{x}_t})\right)^\top \frac{\partial \mathbf{p}_{t,\theta}}{\partial \theta} \tag{8}$$

$$= \sum_{t=1}^{T} \sum_{x \in V} \langle \mathbf{e}_x, \frac{\partial \mathcal{V}_G}{\partial \mathbf{e}}(\mathbf{e}_{\hat{x}_t}) \rangle \frac{\partial p(x|\hat{\mathbf{x}}_{1:t-1})}{\partial \theta} \tag{9}$$

where $\hat{x}_t$ is the selected token at time-step $t$, $T$ is the maximum time-step and $V$ is the given vocabulary. The first term of Equation 8 can be derived from Equation 7 by applying the chain rule.

## 3.2 RE-CENTERED ESTIMATOR

We turn our attention to the first term of Equation 9:

$$\langle \mathbf{e}_x, \frac{\partial \mathcal{V}_G}{\partial \mathbf{e}}(\mathbf{e}_{\hat{x}_t}) \rangle \tag{10}$$

Denoting it as $\delta_{\hat{x}_t \to x}[\mathcal{V}_G]$, Equation 9 becomes

$$\hat{g}_\theta = \sum_{t=1}^{T} \sum_{x \in V} \delta_{\hat{x}_t \to x}[\mathcal{V}_G] \frac{\partial p_\theta(x|\hat{\mathbf{x}}_{1:t-1})}{\partial \theta}. \tag{11}$$

The coefficient $\delta_{\hat{x}_t \to x}[\mathcal{V}_G]$ can be considered as an *affinity factor* indicating how much the discriminator is 'attracted' to the token $x$.

Recall that the gradient $\frac{\partial \mathcal{V}_G}{\partial \mathbf{e}}(\mathbf{e}_{\hat{x}})$ has the direction of greatest increase of the objective $\mathcal{V}_G$ at $\mathbf{e}_{\hat{x}}$. Since this direction is *relative to* $\mathbf{e}_{\hat{x}}$, we suggest taking the inner product with the vector $\mathbf{e}_x - \mathbf{e}_{\hat{x}}$ which has been re-centered relative to $\mathbf{e}_{\hat{x}}$ (Figure 2).

$$\langle \mathbf{e}_x - \mathbf{e}_{\hat{x}_t}, \frac{\partial \mathcal{V}_G}{\partial \mathbf{e}}(\mathbf{e}_{\hat{x}_t}) \rangle \tag{12}$$

This inner product approximates the increase of $\mathcal{V}_G$ when $\hat{x}_t$ is replaced by $x$.

Because this approximation only holds in the neighborhood of $\mathbf{e}_{\hat{x}_t}$, $|\mathbf{e}_x - \mathbf{e}_{\hat{x}_t}| \ll 1$, we also add a kernel to dampen the affinity factor for $\mathbf{e}_x$ that are far away. That is,

$$\delta_{\hat{x}_t \to x}^{RC}[\mathcal{V}_G] = K(\mathbf{e}_x, \mathbf{e}_{\hat{x}_t}) \langle \mathbf{e}_x - \mathbf{e}_{\hat{x}_t}, \frac{\partial \mathcal{V}_G}{\partial \mathbf{e}}(\mathbf{e}_{\hat{x}_t}) \rangle \tag{13}$$

where

$$K(\mathbf{u}, \mathbf{v}) = \frac{1}{\sqrt{1 + |\mathbf{u} - \mathbf{v}|^2}}. \tag{14}$$

See Appendix B for more details. As a result, Equation 9 becomes

$$\hat{g}_\theta^{RC} = \sum_{t=1}^{T} \sum_{x \in V} \delta_{\hat{x}_t \to x}^{RC}[\mathcal{V}_G] \frac{\partial p(x|\hat{\mathbf{x}}_{1:t-1})}{\partial \theta}. \tag{15}$$

In score-function based approaches, the generator is only given feedback on the tokens it samples, so it requires a large amount of trial-and-error in order to figure out how to write realistic sentences.

To use an analogy, imagine a captain at sea (embedding space) searching for treasure. Each iteration of the game, the captain chooses random locations in the space and receives feedback regarding the choices. Score-function based approaches tell us how good the location is but say nothing about how to improve it. In contrast, our method, gives the captain a navigational direction to the next best location. Even if the direction is biased, it should allow the captain to find the treasures with fewer iterations.

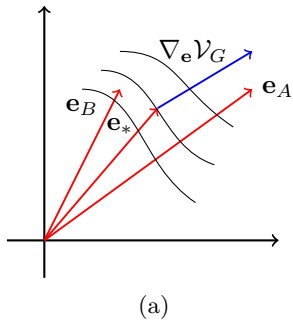 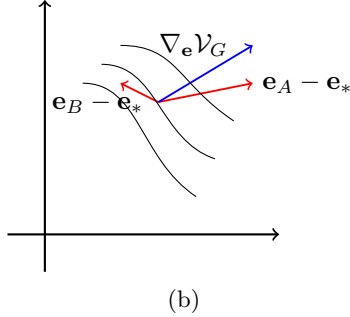

(a)                    (b)

Figure 2: Conceptual contour graph for objective $\mathcal{V}_G$ in embedding space. $\nabla_{\mathbf{e}}\mathcal{V}_G$ is the gradient of $\mathcal{V}_G$ at the sampled token $*$. $B$ is a nearby token with a lower $\mathcal{V}_G$. Affinity factors for each token are calculated as the inner product between the red vectors and the blue vector.
(a) The inner products of $A$ and $B$ with $\nabla_{\mathbf{e}}\mathcal{V}_G$ are both positive even though $B$ leads to a lower $\mathcal{V}_G$. (b) After applying re-centering in the inner product, the affinity factors $\delta^{RC}_{*\to B}[\mathcal{V}_G]$, $\delta^{RC}_{*\to A}[\mathcal{V}_G]$ have opposite signs, and thus better at reflecting the relative suitability of A and B as judged by the discriminator.

## 4   EVALUATION METRICS

We evaluate our model's ability to generate realistic and diverse texts with BLEU-based metric, Fréchet Embedding Distance and language model based metric.

### 4.1   BLEU AND SELF-BLEU

BLEU (Papineni et al., 2002) and Self-BLEU (Zhu et al., 2018) capture text quality and diversity respectively. BLEU, a modified form of n-gram precision, measures local consistency between a set of reference and a set of candidate texts. Self-BLEU is a version of BLEU in which both the reference and candidate texts are drawn from the same corpus. A high Self-BLEU means that members within the corpus are too highly correlated, indicating a lack of diversity.

### 4.2   FRÉCHET EMBEDDING DISTANCE

Fréchet Embedding Distance (FED) (de Masson d'Autume et al., 2019) measures sentimental similarity, global consistency and diversity between reference and candidate texts.

Semeniuta et al. (2018) claims that Fréchet Distance (FD) is not sensitive to the choice of embedding model. However, we notice significant discrepancies between FD scores calculated using different embedding models. The details are discussed in Appendix G.

### 4.3   LANGUAGE MODEL SCORING METHODS

Following Caccia et al. (2018); Zhao et al. (2017), we also evaluate the quality and the diversity of generated samples by Language Model Perplexity (LM) and Reverse Language Model Perplexity (RLM).

**Language Model Score (LM)**. We use a language model trained on real data to estimate the *negative log likelihood per word* of generated sentences. Samples which the language model considers improper, such as ungrammatical or nonsense sentences, have high perplexity.

**Reverse Language Model Score (RLM)**. If we instead train a language model on generated data while evaluating on real data, we can judge the diversity of the generated samples. Low-diversity generated data leads to an overfitted model which generalizes poorly on real data, as indicated by a high perplexity.

## 5 Results & Discussion

In this section, we use two datasets to evaluate CaptainGAN: COCO Image Captions[1] and EMNLP 2017 News [2]. We discuss our results from different perspectives via the evaluation metrics presented in the last section. Furthermore, an ablation study is performed in order to know the contribution of each technique. More details about our experiment setup, datasets, architecture and training techniques such as temperature control are described in Appendix D.

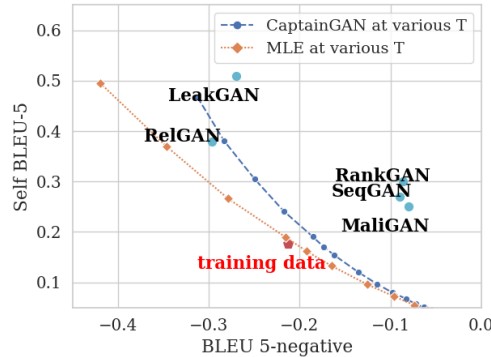 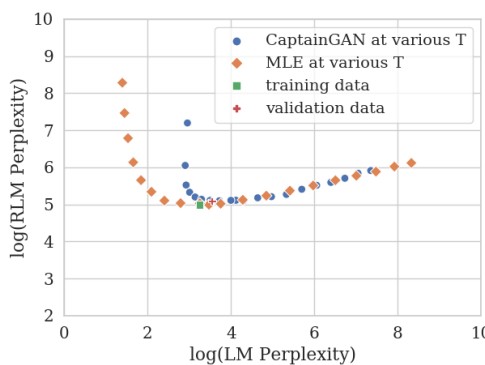

(a) Negative BLEU-5 versus Self-BLEU-5.      (b) Language- and reverse language-model scores.

Figure 3: BLEU scores (left) and language model scores on (right) on EMNLP News. Results for LeakGAN(Guo et al., 2017), MaliGAN(Che et al., 2017), RankGAN(Lin et al., 2017) and SeqGAN(Yu et al., 2017) are from Caccia et al. (2018).

### 5.1 Quality and Diversity

On metrics of local text consistency and diversity, CaptainGAN significantly outperforms prior GAN-based models which rely heavily on pre-training, with the exception of RelGAN. The BLEU/Self-BLEU temperature sweeps shown in Figure 3a for CaptainGAN indicate that CaptainGAN approaches the performance of a language model. On the other hand, in Figure 3b, the language model scores show similar results for CaptainGAN and MLE model. Results for COCO are shown in Appendix E.

### 5.2 Temperature Sensitivity

CaptainGAN is robust to temperature variations. This is demonstrated in Figure 4a, which depicts how the softmax temperature influences the FED score. Regardless of model, changes in softmax temperature are certain to affect the FED, however, CaptainGAN shows the least variation.

### 5.3 Global Consistency

CaptainGAN improves global consistency comparing to prior works and language model on LM perplexity and FED score, respectively. As shown in Table 1, our method significantly reduces the gap of perplexity between GAN-based method and language model, which is directly trained to minimize the perplexity. Besides, we show that CaptainGAN outperforms MLE trained model on FED score as shown in Table 1.

---

[1]The preprocessed COCO dataset is available at `https://github.com/pclucas14/GansFallingShort/tree/master/real_data_experiments/data/coco`

[2]The preprocessed EMNLP dataset is available at `https://github.com/pclucas14/GansFallingShort/tree/master/real_data_experiments/data/news`

Table 1: FED & LM scores on EMNLP 2017 News

| Model | Train FED | Val. FED | LM Score |
|---|---|---|---|
| Training data | 0.0050 | 0.0120 | 3.22 |
| MLE | **0.0100** | 0.0194 | **3.43** |
| SeqGAN | 0.1234 | 0.1422 | 6.09 |
| MaliGAN | 0.1280 | 0.1504 | 6.30 |
| RankGAN | 0.1418 | 0.1431 | 5.76 |
| LeakGAN | 0.0718 | 0.0691 | 4.90 |
| RelGAN | 0.0462 | 0.0408 | 3.49 |
| CaptainGAN | 0.0120 | **0.0184** | 4.04 |

Note: All the GANs, except CaptainGAN, are pretrained by MLE.

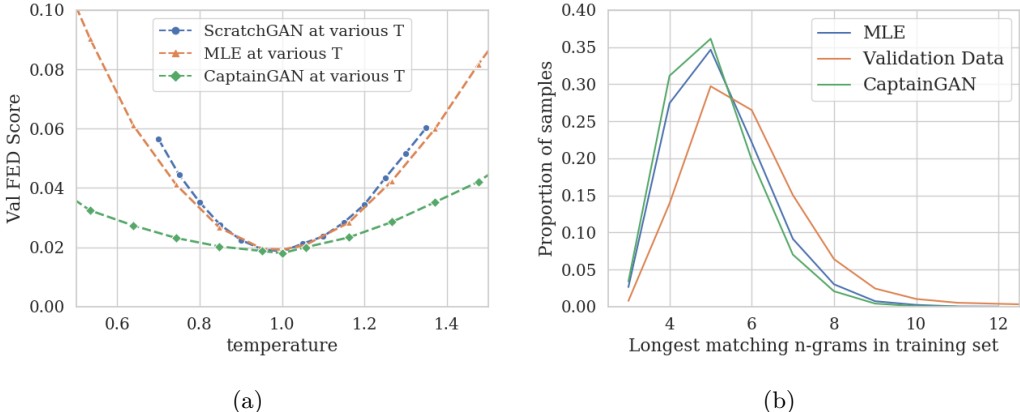

(a)                                                        (b)

Figure 4: (a) The influence of different temperature values on FED scores measured on EMNLP 2017 News. (b) Longest matching n-grams on EMNLP 2017 News training data.

## 5.4 Matching N-grams of Training Data

Text generating models are often criticized for simply memorizing texts seen during training instead of learning how to model the true data distribution. We show the longest matching n-grams between text from the training data and samples in Figure 4b. CaptainGAN generates more samples with small matching n-grams (n < 5) comparing to MLE model, implying that our model can generate novel sentences without resorting to memorizing phrases from the training data.

## 5.5 CaptainGan As A Language Model

CaptainGAN's strong performance on the evaluation metrics described earlier shows that it is capable of generating realistic text. Since the generator in our case is a RNN which can produce conditional probabilities at each timestep, this raises the question of whether or not an explicit language model is being learned.

We feed real data to our generator and calculate the perplexity to measure the generator's suitability as a language model. Surprisingly, as shown in Table 2, the perplexity of the generator on both training and validation data is unusually high. A similar phenomenon was noted in de Masson d'Autume et al. (2019), though not as extreme.

We explain this phenomenon as follows: MLE models are trained specifically for low perplexity, since they minimize KL divergence. However, CaptainGAN minimizes alternative loss (Equation 2), which is similar to minimizing reverse KL divergence (Arjovsky & Bottou,

2017). This objective assigns an extremely low cost to mode dropping and does not force the generator to mimic all aspects of real data. This can result in poor modeling of the likelihood, but does not necessarily lead to poor sample generation. (Theis et al., 2016; Hashimoto et al., 2019).

Following 5.4, we conclude that our model's ability to generate realistic text cannot be the result of simply plagiarizing training samples.

Table 2: Word level perplexity on EMNLP 2017 News.

| Model | Train Perp. | Val. Perp. |
|---|---|---|
| Random. | 5.3k | 5.3k |
| MLE | 26 | 34 |
| CaptainGAN | 14.9k | 16.8k |

Note: Perplexity is calculated as geometric mean over sentences.

### 5.6 Comparison to score function-based methods

Our method is able to outperform score function-based approaches using small batch sizes. Score function-based approaches often rely on large batch sizes for variance reduction (de Masson d'Autume et al., 2019), but this does not appear to be necessary for our re-centered estimator. This confirms the reasoning at the end of Section 3.2, and greatly reduces our memory requirements during training.

In addition, since gradients are naturally calculated for every timestep, there is no need for Monte-Carlo estimation of rewards like in score function-based approaches. Because our generator receives a richer learning signal, we use a 5:1 ratio for discriminator-generator updating. This greatly lessens our computational requirements.

### 5.7 Ablation Study

To understand the impact of each component of the CaptainGAN, we conduct an ablation study. In Table 3, we show the influence of adding new features, namely spectral normalization, re-center estimator, pretrained embeddings and discriminator regularization decoupled weight decay (Loshchilov & Hutter, 2018) and one-sided label smoothing (Salimans et al., 2016), by scoring FED on validation data.

We observe that:

- As shown in Zhou et al. (2019), an unrestricted objective leads to an uninformative gradient. Without spectral normalization, the Lipschitz constant of the discriminator is unbounded making it difficult to update the generator with a gradient-based approach.

- Rather unexpectedly, pretrained embeddings do not lead to any significant improvement, which is surprising given our method's dependence on the discriminator's embedding space. The embeddings which the discriminator learns from scratch are capable of giving the generator an effective learning signal.

- The application of the re-centered estimator is responsible for a 20% improvement in FED, confirming its effectiveness.

- Equation 14 is dependent on the norm of the embeddings, which explains the effectiveness of discriminator weight regularization at improving FED.

Table 3: Ablation study on EMNLP 2017 News validation data

| Model | Val. FED |
|---|---|
| straight through estimator | 0.383 |
| + spectral normalization | 0.047 |
| + re-center estimator | 0.037 |
| + pre-trained embeddings | 0.035 |
| + discriminator regularization | 0.018 |

## 6 Conclusion and Future Work

In this work we have presented CaptainGAN, an effective gradient-based method for quickly training a text generating GAN without pretraining. Starting from the straight-through estimator, we derive a re-centered gradient estimator that improves the quality/diversity of generated texts as measured by various standard metrics. In future work, we plan to investigate our estimator in more theoretical detail and further decrease its bias.

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

## APPENDIX A    NOTATION

| Symbol | Meaning |
|---|---|
| $V = \{x_1, \ldots, x_v\}$ | a predefined vocabulary |
| $x$ | a token belongs to $V$ |
| $\hat{x}$ | a token sampled from $V$ |
| $\mathbf{x}$ | a sequence of tokens belong to $V$ |
| $\hat{\mathbf{x}}$ | a sequence of tokens sampled from $V$ |
| $\top$ | a transpose operation |

## APPENDIX B    THE IMPORTANCE OF KERNEL

To verify the necessity of the kernel term (Equation 14), we directly apply Equation 12 to Equation 11 as follows:

$$
\sum_{x \in V} \langle \mathbf{e}_x - \mathbf{e}_{\hat{x}}, \frac{\partial \mathcal{V}_G}{\partial \mathbf{e}}(\mathbf{e}_{\hat{x}}) \rangle \frac{\partial p_\theta(x)}{\partial \theta}
$$

$$
= \sum_{x \in V} \langle \mathbf{e}_x, \frac{\partial \mathcal{V}_G}{\partial \mathbf{e}}(\mathbf{e}_{\hat{x}}) \rangle \frac{\partial p_\theta(x)}{\partial \theta}
$$

$$
- \sum_{x \in V} \langle \mathbf{e}_{\hat{x}}, \frac{\partial \mathcal{V}_G}{\partial \mathbf{e}}(\mathbf{e}_{\hat{x}}) \rangle \frac{\partial p_\theta(x)}{\partial \theta} \tag{16}
$$

where $\hat{x}$ is the selected token and $x$ is a tokens in the vocabulary V.

Let $f(x) = \langle \mathbf{e}_x, \frac{\partial \mathcal{V}_G}{\partial \mathbf{e}}(\mathbf{e}_{\hat{x}}) \rangle$ and $b = \langle \mathbf{e}_{\hat{x}}, \frac{\partial \mathcal{V}_G}{\partial \mathbf{e}}(\mathbf{e}_{\hat{x}}) \rangle$. $b$ does not depend on $x$ and is constant in the summation.

$$
\sum_{x \in V} (f(x) + b) \frac{\partial p_\theta(x)}{\partial \theta}
$$

$$
= \frac{\partial}{\partial \theta} [\sum_{x \in V} (f(x) + b) p_\theta(x)]
$$

$$
= \frac{\partial}{\partial \theta} [(\sum_{x \in V} f(x) p_\theta(x)) + b (\sum_{x \in V} p_\theta(x))]
$$

$$
= \frac{\partial}{\partial \theta} [(\sum_{x \in V} f(x) p_\theta(x)) + b * 1]
$$

$$
= \frac{\partial}{\partial \theta} \sum_{x \in V} f(x) p_\theta(x) \tag{17}
$$

Without the kernel, the update to $\theta$ will be the same as prior work and thus carries the same drawback.

## APPENDIX C    MODEL STRUCTURE

We use a pre-trained word embedding of 300 dimensions to initialize all embedding weights. The embeddings are pre-trained using the fastText library Bojanowski et al. (2016) on the corresponding training data of each task.

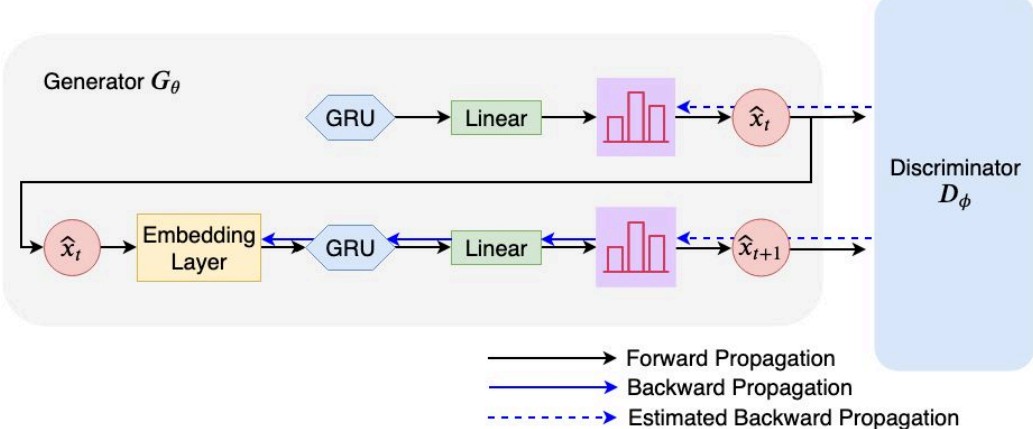

Figure 5: Model architecture and computation flow

| embedding lookup |
| --- |
| GRU 1024 |
| dense $d_E$ |
| dense V |

Table 4: Generator structure, $d_E$ is the embedding size, and V is the vocabulary size.

## C.1 GENERATOR

The purpose of the first dense layer is to project the GRU cell outputs into the embedding space. The second dense layer converts the projected embedding vectors to logits over the tokens and is initialized as the transpose of the embedding weights.

At the first timestep, the input to the embedding layer is the start-of-sentence token. The vocabulary also contains an end-of-sentence token. If the generator outputs this token at any timestep, the sentence ends and its length is set to this timestep.

## C.2 DISCRIMINATOR

| embedding lookup |
| --- |
| conv3-512 relu |
| conv4-512 relu |
| avg pool2 |
| conv3-1024 relu |
| conv4-1024 relu |
| global average pool |
| dense 1024 relu |
| dense 1 |

Table 5: Discriminator, all conv & pooling layers use 'same' padding.

To perform consistent convolution on sentences of different lengths, we use a special masking mechanism on all layers. For all convolution and pooling layers, we mask out input features where the receptive field is out-of-bounds with respect to the unpadded input sentence.

## Appendix D  Experimental details

### D.1  Setup

#### D.1.1  Datasets

We use two datasets COCO Image Caption dataset Chen et al. (2015) and EMNLP 2017 News dataset[3]. Results are reported on EMNLP 2017 News if not specified. For each datasets, 10k sentences are set aside as validation data.

**COCO Image Captions.** Sentences are limited in length to 24 tokens. The vocabulary size is 4.6k tokens. Training data consists of 10k sentences[4].

**EMNLP 2017 News.** Sentences in this dataset are much longer and more complicated, with a maximum length of 50 tokens and a vocabulary of 5.7k words. Training data consists of 300k sentences. [5]

#### D.1.2  Architecture and Techniques

Our architecture is described in Appendix C. Spectral normalization of the discriminator's weights Miyato et al. (2018) is critical to our method as without it the gradients are too unstable and convergence becomes impossible (Section 5.7).

#### D.1.3  Temperature Control

Following Caccia et al. (2018), we adjust the softmax temperature parameter at sampling time to measure the trade-off between quality and diversity. Increasing the temperature lowers the differences between softmax probabilities, which leads to diverse but low-quality samples. On the other hand, reducing the temperature leads to high-quality yet low-diversity samples.

### D.2  training details

The discriminator and generator updates are performed with a 5:1 ratio. We use Adam Kingma & Ba (2014) with learning rate $= 5.0 \cdot 10^{-5}$, $\beta_1 = 0.5$, and $\beta_2 = 0.999$. Decoupled weight decay regularization Loshchilov & Hutter (2018) is applied to all variables in the discriminator, and $\lambda = 0.05 \cdot$ learning rate. All our models are trained using an NVIDIA GTX 1080 Ti.

To evaluate samples on BLEU-5, LM versus RLM, and FED, we use 10k validation sentences as reference texts for both COCO and EMNLP 2017 News. For Self-BLEU, we randomly select half of the 10k samples as reference texts and leave the remainder as target texts. A smoothing function is used on BLEU-based metrics. See Appendix D.4 for more details. The maximum likelihood model architecture in this paper follows the setting of de Masson d'Autume et al. (2019). We could not reproduce the exact results reported in de Masson d'Autume et al. (2019), which we believe stems from discrepancies between our trained language model and theirs.

### D.3  One-sided label smoothing

One-sided label smoothing has been shown to reduce the vulnerability of neural networks to adversarial examples and it is strongly recommended for GAN training Salimans et al. (2016). Therefore, the first term of Equation 1 is reformulated as

$$\mathbb{E}_{\mathbf{x} \sim p_{data}}[\alpha \log \mathcal{D}_\phi(\mathbf{x}) + (1 - \alpha) \log(1 - \mathcal{D}_\phi(\mathbf{x}))] \tag{18}$$

where $\alpha = 0.9$ in our experiments.

---

[3]`http://www.statmt.org/wmt17/`

[4]The preprocessed COCO dataset is available at `https://github.com/pclucas14/GansFallingShort/tree/master/real_data_experiments/data/coco`

[5]The preprocessed EMNLP dataset is available at `https://github.com/pclucas14/GansFallingShort/tree/master/real_data_experiments/data/news`

## D.4 BLEU SMOOTHING

One of the issue with BLEU is that in the case that a higher order n-gram precision of a sentence is 0, then the BLEU score will be 0, resulting in severely underestimation. This is due to the fact that BLEU is calculated by the geometric mean of precision. To solve this, we applied the smoothing technique as follows:

$$m_n^{'} = \epsilon \text{ if } m_n = 0 \tag{19}$$

where $m_n$ is the original match count, $m_n^{'}$ is the one used during BLEU calculation. We've picked the smoothing factor $\epsilon = 0.1$ as proposed by Chen & Cherry (2014).

## APPENDIX E    COCO RESULTS

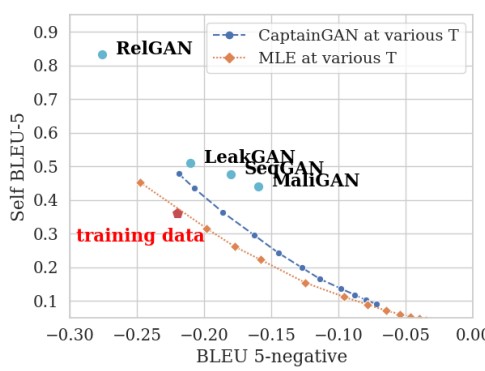
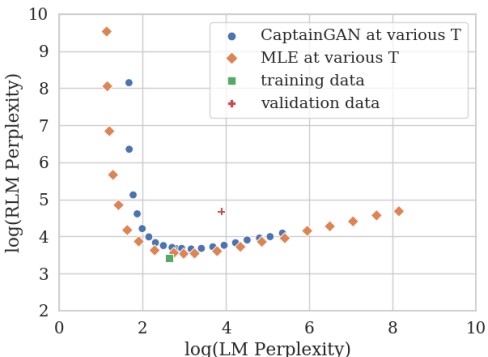

(a) Negative BLEU-5 versus Self-BLEU-5.

(b) Language- and reverse language-model scores.

Figure 6: BLEU scores (left) and language model scores on (right) on COCO. Results for LeakGAN(Guo et al., 2017), MaliGAN(Che et al., 2017), RankGAN(Lin et al., 2017) and SeqGAN(Yu et al., 2017) are from Caccia et al. (2018).

See Figure 6.

## APPENDIX F    SENTENCE LENGTH CONSISTENCY

The sentence length of samples which are generated by CaptainGAN is highly related to the training data, as shown in Figure 7. This suggests that CaptainGAN is able to fully learn the distribution of sentence length by gradient estimator without additional mechanism like positional encoding or dense reward.

## APPENDIX G    DETAILS ON FRÉCHET EMBEDDING DISTANCE

Calculation of the FED score depends on the Universal Sentence EncoderCer et al. (2018). However, the version of Universal Sentence Encoder[6] used in de Masson d'Autume et al. (2019) is incompatible with TensorflowAbadi et al. (2015) version 1.7 or newer. Furthermore, it would be infeasible to limit the experimentation environment to such ancient version of Tensorflow. Therefore, as a work around, we decide to report our FED score, as shown in Table 6, with Universal Sentence Encoder Large[7] instead, which is compatible with the current version of Tensorflow (1.14 as of Jul 2019).

While there are many benefits to using the FED metric, it is not without drawback. A drawback that we've observed is that the FED score will be significantly underestimated

---

[6]https://tfhub.dev/google/universal-sentence-encoder/2
[7]https://tfhub.dev/google/universal-sentence-encoder-large/3

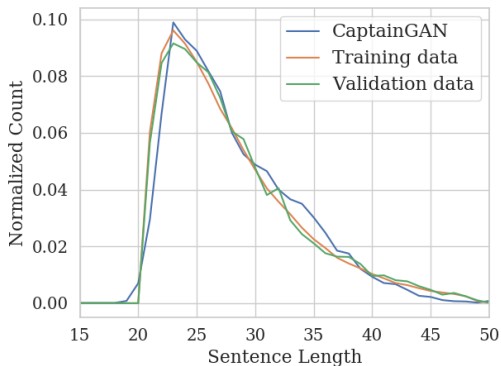

Figure 7: Normalized counts of different sentence length on EMNLP 2017 News.

Table 6: FED scores on EMNLP 2017 News based on Universal Sentence Encoder Large

| Model | Train FED | Val. FED |
| --- | --- | --- |
| Training data | 0.0059 | 0.0136 |
| MLE | 0.0130 | 0.0244 |
| SeqGAN | 0.1333 | 0.1414 |
| MaliGAN | 0.1392 | 0.1412 |
| RankGAN | 0.1554 | 0.1474 |
| LeakGAN | 0.0884 | 0.0878 |
| RelGAN | 0.0504 | 0.0514 |
| CaptainGAN | 0.0164 | 0.0232 |

whenever we are using a small number of samples, as shown in Figure 8. Therefore, we've ensured that sufficient samples are used in our evaluation.

## APPENDIX H   GENERATED SAMPLES

Training samples of COCO and EMNLP 2017 News can be found in Table 7. Randomly picked samples from MLE model and CaptainGAN are available in Table 8 and Table 9.

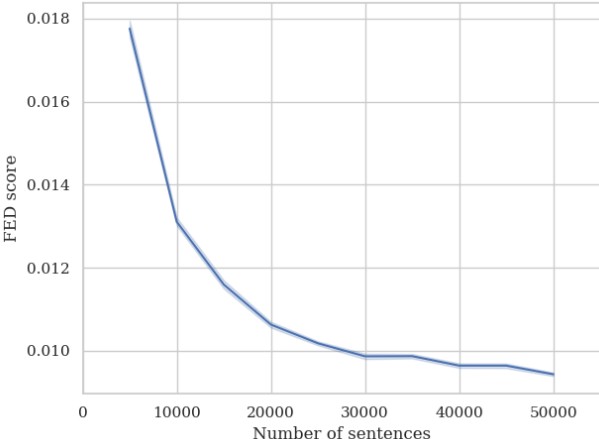

Figure 8: FED scores on different sampling size of training data and CaptainGAN samples.

| COCO |
| --- |
| - a person standing in a boat on the water . |
| - three sheep are grazing on the city sidewalk . |
| - kenya airways airplane wing , engine and cabin on the tarmac . |
| - light poles in the snow with yellow traffic lights mounted on them . |
| - we are looking at the floor between the toilet and the wall . |
| - someone taking a photo of a small residential bathroom . |
| - a cat sits on the seat of a bicycle and looks down at another cat on a snowy day . |
| - large set of motorcycles all lined up down a street . |
| **EMNLP 2017 News** |
| - apple clearly doesn ' t want to see the dollar value of its u . k . earnings fall by a similar amount . |
| - we do look at the number of hours we produce , and measure that against the religious make - up of society . |
| - the u . s . has struggled to find an effective ground force to take on isis in syria , where president obama has ruled out a u . s . ground combat role . |
| - it is a happy and kind world that we live in on this show and that is where i hope we can live in real life . |
| - while men and women may end up earning roughly the same amount in the same jobs , men are more likely to end up in higher - paying roles in the tech industry . |
| - but while they were beaten by a better side , the tie did reveal what i think has been city ' s biggest problem this season : they have lost the ability to score against good teams . |
| - according to facebook ' s policies , accounts can be suspended if law enforcement believe individuals are at risk of harm . |
| - she has to take personal responsibility for this - when she was health secretary she was told by those who know best that her decision to cut student places would have a damaging impact . |

Table 7: Training samples on EMNLP 2017 News and COCO dataset.

**COCO**

- the bathroom bowl with focus on the vanity .
- a man racer in a car brushing a carved cake .
- a lady riding a motorcycle and a street while sitting on a pedestal track .
- a commercial man are standing in the bicycle next to the back of an orange computer .
- an airplane the bus is headed as something in the sky .
- people riding on a bike on a surfer surfing doing the ocean .
- a plane propped up in front of a vehicle .
- a man wearing a cat in front of a motorcycle .

**EMNLP 2017 News**

- a one - game event and a limited number this dropped by 38 per cent to find out what we could want , to be as much done as we finish as the games .
- more than 30 per cent of calais camps at risk were taken between children and child refugees crossing the mediterranean to fall apart .
- the uk government got on the deal when it came to other eu countries that allowed workplace checks for 26 .
- so women i didn ' t want my parents to work hard with me at the time of the task of making it presents .
- black voters feel there may not be enough choice between former president george w . bush and i have to go along with problems and support the people that are in a swing state for president .
- he sold out to its eighth list , but a stunning showing that what cost the ability to start the league failed ?
- let ' s catch him down his back phone one morning john f kennedy , where he is in a us meeting group with supporters of love not being successful in the fields .
- then there ' s a couple of chances that they won competition - - the result from what is of me going out there that i really will play for the foundation .

Table 8: Randomly selected MLE samples on EMNLP 2017 News and COCO with temperature = 1.0.

**COCO**

- a black motorcycle parked next to a river .
- a bathroom with a mirror , a shower and a toilet .
- two people sit on a building in front of a building .
- a group of people walking around the corner of a building .
- people are riding bikes on a city street .
- there is a picture of a bathroom in a bathroom .
- an empty city street with cars parked on the side of the road .
- a man taking a picture of a white bathroom .

**EMNLP 2017 News**

- if i go to the party , it ' s a disaster , and i ' m going to keep it right .
- i think i ' ve had some great strength in my life and i want that to be here for me .
- obama pointed to reporters claiming he was willing to discuss the situation , but the u . s . have been military concern by american troops , including a nuclear deal in march .
- a 16 - year - old man was arrested after the death of a man shot and arrested him on suspicion on his behalf .
- the problem is that they are hard to find out what they want to see where they are and what they can achieve their past two years .
- now it ' s important to me and i am thinking so hard to make it the best for a long time .
- great britain is suggesting the u . s . - led coalition figures at the australian foreign investor have raised more than $ 250 , 000 in the construction of operations .
- it is a simple way to believe that if it can happen , then i ' m not the right choice .

Table 9: Randomly selected CaptainGAN samples on EMNLP 2017 News and COCO with temperature = 1.0.

