# OpenReview forum: "CaptainGAN: Navigate Through Embedding Space For Better Text Generation"
_ICLR.cc/2020/Conference — Reject_

### Official Review · AnonReviewer3 · 2019-10-22
**Official Blind Review #3**

**Rating:** 6

**Review:**

The authors propose CaptainGAN, a method using the straight-through gradient estimator to improve training of the generator for text generation.

The paper is well-written and the evaluation seems thorough, comparing to relevant baselines.

Comments:

Figure 3: the caption refers to Caccia et al. for results on LeakGAN, MaliGAN and seqGAN, but unless I’ve missed it, RelGAN hasn’t yet been introduced by name as a baseline? The citation is given in the opening part of the introduction, in an enumeration, but isn’t revisited later in the text - not even here where the results of the model are introduced. Given that it seems, according to the presented results, to be the most competitive of the GAN models that the authors are comparing to, maybe it’s worth adding more contextual information on RelGAN to the Background section?

For their method, the authors should report an average performance over several random seeds and provide the standard deviation / confidence intervals, for the readers to be able to assess the stability of the method and the significance of the improvement reported in the results.

I find Section 5.5. particularly interesting, as well as the reported perplexity in Table 2. The authors provide 3 bullet points to explain the unusually high perplexity of the generator on the training and validation data. I feel that the explanations that are given are at the moment vague and not visibly backed by data, therefore being speculative. Obviously, point 1) is hard to quantify - but point 2) could possibly be at least partially quantified - if the hypothesis is that names, places, punctuation marks etc play an important role in the reported perplexity score, then maybe the authors could test this by correlating model perplexity on sentences with whether those sentences contain these types of words?


**Experience Assessment:**

I have read many papers in this area.

**Review Assessment: Checking Correctness Of Derivations And Theory:**

I assessed the sensibility of the derivations and theory.

**Review Assessment: Checking Correctness Of Experiments:**

I carefully checked the experiments.

**Review Assessment: Thoroughness In Paper Reading:**

I read the paper at least twice and used my best judgement in assessing the paper.

---

> ### Author Response · Authors · 2019-11-13
> **Reply to Reviewer #3**
>
> Regarding the report of an average performance with confidence interval using different random seeds:
>
> Thank you for the suggestion. Providing average performance and confidence intervals is important. However, due to time constraints, we will not be able to add the results before 11/15. Instead, we will try to include the results in our final submission.

---

> > ### Comment · AnonReviewer3 · 2019-11-13
> > **Regarding multiple random seeds**
> >
> > Thank you, please include if possible.
> > It would strengthen the evaluation and the results.
> >
> > Are there any plans of backing some of 5.5 with additional experiments? (see comment above)

---

> > > ### Author Response · Authors · 2019-11-15
> > > **Reply to Reviewer #3 about section 5.5**
> > >
> > > Sorry for the late reply.
> > >
> > > We've rewritten the section 5.5 and have a response at https://openreview.net/forum?id=H1gy1erYDH&noteId=HkeF9zh9jS .

---

> ### Author Response · Authors · 2019-11-14
> **Reply to Reviewer #3**
>
> Regarding the lack of information about RelGAN:
>
> RelGAN is a GAN architecture using the Gumbel-Softmax estimator. In our revision, we’ve added a subsection (2.1 Continuous Relaxations) to explain the drawback of using the Gumbel-Softmax estimator. Also, we’ve added RelGAN scores to Figure 3 and Figure 6. We show that CaptainGAN is competitive with RelGAN (without additional pretraining before adversarial training) in terms of Bleu/SelfBleu and outperforms RelGAN’s FED score.

---

> ### Author Response · Authors · 2019-11-14
> **Reply to Reviewer #3**
>
> Regarding the explanation of unusually high perplexity:
>
> We’ve rewritten Section 5.5 to better explain the unusually high perplexity. The main reason is CaptainGAN minimizes different objective from MLE models. This objective assigns an extremely low cost to mode dropping and does not force the generator to mimic all aspects of real data. This can result in poor modeling of the likelihood but does not necessarily lead to poor sample generation. For more details, please see the revised version of section 5.5. Moreover, we are planning to add more experiments to measure the severity of mode dropping. Due to the time constraint, we will report the result at the final submission.

---

### Official Review · AnonReviewer1 · 2019-10-23
**Official Blind Review #1**

**Rating:** 6

**Review:**

This paper attempts to solve the problem of non-differentiable connection between the generation and discriminator of a GAN. The authors come up with an estimator of the gradient for the generator from the gradient of the discriminator, which was disconnected previously. With this change, the model should be able to  select better tokens than random selection, which could leads to more robust training. The experiment results on both COCO Image Captions and EMNLP 2017 News datasets justify the authors' argument.

**Experience Assessment:**

I do not know much about this area.

**Review Assessment: Checking Correctness Of Derivations And Theory:**

I did not assess the derivations or theory.

**Review Assessment: Checking Correctness Of Experiments:**

I did not assess the experiments.

**Review Assessment: Thoroughness In Paper Reading:**

I made a quick assessment of this paper.

---

### Official Review · AnonReviewer2 · 2019-11-01
**Official Blind Review #2**

**Rating:** 3

**Review:**

The submission proposes to train a GAN on discrete sequences using the straight-through Gumbel estimator introduced in Jang et al. (2016) in combination with gradient centering. The proposed approach is evaluated on COCO and EMNLP News in terms of BLEU and Self-BLEU scores, Fréchet Embedding Distance, Language Model Score, and Reverse Language Model Score.

My assessment is that the submission is below the acceptance bar, mainly due to clarity and novelty concerns. The proposed approach does have empirical backing, but I would argue that it is a very straightforward application of the straight-through Gumbel estimator to GANs, which is itself similar to existing work on applying the Gumbel-softmax estimator to GANs (Kusner & Hernández-Lobato, 2016). Detailed comments can be found below.

The submission does not feel self-contained. For instance, it borrows notation from Jang et al. (2016) without explicitly acknowledging it, and my personal experience is that reading Jang et al. (2016) beforehand makes a big difference in terms of clarity in Section 2.2.

The notation is inconsistent and confusing, and gets in the way of understanding the proposed approach. Here’s a (non-exhaustive) list of examples:

- The reward function is first introduced as f_\phi(\mathbf{x}) above Equation 3, but all subsequent mentions of the reward function use f_\phi(\hat{\mathbf{x}}).
- The \mathbf{m}_\theta variable is introduced in Equation 5 and is immediately replaced with \mathbf{p}_\theta, which adds notational overhead without any benefit.
- The difference between \hat{\mathbf{x}} and \hat{x} is not explained in the text. From the context I understand that \hat{x} is a categorical scalar in {1, …, V}; is this correct?
- In Equation 6, x_1, …, x_V are used to denote the *values* that \hat{x} can take. This clashes with the previous convention that \mathbf{x} is a sequence sampled from p_{data} (Equation 1). Given that convention and the difference between bolded and non-bolded variables discussed above, I would have expected that x_1, …, x_V would correspond to the categorical values of elements of the \mathbf{x} sequence. That contributes to confusion in Equation 9, where \mathbf{e}_{x_t} and p_\theta(x_t) are *not* time-dependent.
- Equation 8 sums over time steps, but the first summation that appears in Equation 8 does not make use of the temporal index. There is also a symbol collision for T, which is used both as the sequence length and as the "transpose" symbol.

As a result, the proposed centering method and the rationale for it is still not entirely clear to me. In particular, is the gradient centering approach necessary to avoid the drawback of score function-based approaches (i.e. the generator is only given feedback on the tokens it samples), or does the non-centered, straight-through variant of the proposed approach also avoid this drawback?

I’m also not convinced that the centering heuristic is a crucial component of the proposed approach when the biggest improvement observed over the straight-through baseline is obtained by adding spectral normalization. I would argue that the proposed approach is a straightforward application of the straight-through Gumbel gradient estimator to GAN training, which is similar in spirit to work by Kusner & Hernández-Lobato (2016) (not cited in the submission) -- the main difference being that the latter uses the Gumbel-softmax distribution directly and anneals the temperature parameter over the course of training. A comparison between the two would be warranted.

References:

- Kusner, M. J., & Hernández-Lobato, J. M. (2016). GANs for sequences of discrete elements with the Gumbel-softmax distribution. arXiv:1611.04051.

**Experience Assessment:**

I have published one or two papers in this area.

**Review Assessment: Checking Correctness Of Derivations And Theory:**

I assessed the sensibility of the derivations and theory.

**Review Assessment: Checking Correctness Of Experiments:**

I assessed the sensibility of the experiments.

**Review Assessment: Thoroughness In Paper Reading:**

I read the paper at least twice and used my best judgement in assessing the paper.

---

> ### Author Response · Authors · 2019-11-12
> **Reply to Reviewer #2**
>
> We appreciate your rigorous review of our work. We’ve made several revisions with your feedback in mind.
>
> Regarding notation:
> We have added an acknowledgment for the usage of notation from Jang et al. (2016). We have also simplified and clarified the notation as follows:
>
> - V = {x_1, …, x_v} stands for a predefined vocabulary of size v.
> - x stands for a discrete token in V.
> - \hat{x} stands for a discrete token sampled from V by the generator G.
> - \mathbf{x} stands for a sequence of discrete tokens belong to V.
> - \hat{\mathbf{x}} stands for a sequence of discrete tokens sampled from V.
> - \top stands for the transpose operation.
>
> We hope these adjustments improve the clarity of our paper.

---

> ### Author Response · Authors · 2019-11-13
> **Reply to Reviewer #2**
>
> Regarding the question about the difference between the work of Kusner & Hernández-Lobato (2016) and our work:
>
> We’ve added a subsection (2.1 Continuous Relaxations) which explains the difference between the work of Kusner & Hernández-Lobato (2016) and our work. We specifically do not use Gumbel-Softmax Estimator for text generation. As explained in our revision, using the Gumbel-Softmax Estimator leads to a training interaction with potentially pathological quirks which is not reflective of the actual text generation process.
>
> Keeping training and inference consistent (both using sampled tokens) is why we use the Straight-Through estimator (note: not a “Straight-Through Gumbel estimator”). However, that choice by itself doesn’t guarantee good results. Our main contribution is how to apply the straight through estimator in an ideal way so that performance is acceptable, and that requires ensuring that a useful gradient is available during the backward pass.

---

> ### Author Response · Authors · 2019-11-15
> **Reply to Reviewer #2**
>
> Regarding the contribution:
>
> Yes, the straight-through estimator helps avoid the drawback of the score-function estimator by providing extra information, which is the gradient of the discriminator, to the generator from the discriminator. There is technically an infinite number of possible “straight-through estimators”. The one we think of as the straight-through estimator is just the most obvious way to define the backward gradient (by pretending the activation is an identity function) - but there is no reason to think that it is the best. Thus it is worth searching for modifications.
>
> Spectral normalization is not so much an “addon” as it is a crucial prerequisite for our method. Since we want to incorporate the gradient of the discriminator in the generator, we need to bound the Lipschitz constant of the discriminator (using spectral normalization) and makes it possible for the generator to use gradient of the discriminator effectively. For more details, please see the revised version of section 5.7.
>
> Our experiments show that the recentering trick increases 20% FED compared to the baseline straight-through estimator (both using spectral normalization), which is why we feel it is worth incorporating.

---

> > ### Comment · AnonReviewer2 · 2019-11-15
> > **Reply**
> >
> > Thank you for your response.
> >
> > I was mistaken in my understanding that the straight-through estimator proposed in this submission is the same as the Gumbel straight-through estimator introduced in Jang et al. (2016); thank you for clarifying. I now understand that the difference between the two is that the former uses “d/d\theta softmax(logits)”, whereas the latter uses “d/d\theta softmax(logits + gumbel_noise)”; is this correct?
> >
> > I think the newly-added Section 2.1 and your justification for the use of spectral normalization contradict each other. On one hand, Section 2.1 dismisses continuous relaxations on the basis that the difference between a one-hot encoded token and a distribution over tokens is easy to spot by the discriminator, which then becomes very certain of its predictions. On the other hand, the proposed approach requires the discriminator’s Lipschitz constant to be bounded for the generator to use the gradient of the discriminator effectively. It seems to me that this line of reasoning could just as well apply to continuous relaxations, because bounding the Lipschitz constant of the discriminator limits how “certain” it can be of its prediction. Would that not nullify the objection raised in Section 2.1?

---

> > > ### Author Response · Authors · 2019-11-15
> > > **Reply**
> > >
> > > Reply to the first question: Yes.
> > >
> > > The main purpose of constraining Lipschitz constant is to make the gradient informative even if the supports of real samples and generated samples are completely disjoint (or, make the loss surface between two supports smooth). In this case, the discriminator can still be confident about its output.
> > >
> > > We have cited Arjovsky & Bottou (2017) & Zhou et al. (2019). We hope the detailed discussion about gradient vanishing and gradient uninformativeness of these works can help us explain it.
> > >
> > > After that, the main problem with continuous relaxation is that the generator is led to produce spiky outputs since it must force it's distribution over tokens to be like the real data, which is a one-hot encoded token.
> > > Spectral normalization can help continuous relaxations but overall the discriminator will still influence the generator to produce spiky outputs (alternatively, we could let the word embeddings non-trainable to make the input of discriminator be word embeddings instead of probability).
> > >
> > >
> > > References:
> > > - Martín Arjovsky and Léon Bottou. Towards principled methods for training generative adversarial networks. ArXiv, abs/1701.04862, 2017.
> > > - Zhiming Zhou, Jiadong Liang, Yuxuan Song, Lantao Yu, Hongwei Wang, Weinan Zhang, Yong Yu, and Zhihua Zhang. Lipschitz generative adversarial nets, 2019.

---

### Public Comment · ~Dianqi_Li1 · 2019-11-05
**Missing paper reference**

Hi, Thanks for the good work. Just a minor comment: your experiment uses the results of MaliGAN and RankGAN. However, you didn't cite these two papers in your reference.

---

> ### Author Response · Authors · 2019-11-12
> **Reply to Dianqi Li**
>
> Thank you for the friendly reminder. We've uploaded a revision with added citations for RankGAN and MaliGAN.

---

### Author Response · Authors · 2019-11-15
**Summary of major changes**

Summary of major changes

We thank all the reviewers for their insightful comments. Your suggestions have helped us to make important revisions to our paper. Major changes are as follows:

- A table of notation has been added in Appendix A.
- A new section (2.1 - Continuous Relaxations) has been added. We hope it is helpful to demonstrate the difference between approaches to non-differentiability.
- Section 5.5 has been rewritten to better explain the unusually high perplexity in response to Reviewer #3.
- Section 5.7 has been revised to clarify the contribution of our work.

Additions that will be made in the final submission:
- error-bar: More experiments will be conducted (using different random seeds) for providing an average performance and confidence intervals for CaptainGAN.
- perplexity distribution: A perplexity distribution will be plotted for showing the severity of mode dropping.

---

### Decision · Program_Chairs · 2019-12-19

**Decision:**

Reject

**Comment:**

This paper proposes a method to train generative adversarial nets for text generation. The paper proposes to address the challenge of discrete sequences using straight-through and gradient centering. The reviewers found that the results on COCO Image Captions and EMNLP 2017 News were interesting. However, this paper is borderline because it does not sufficiently motivate one of its key contributions: the gradient centering. The paper establishes that it provides an improvement in ablation, but more in-depth analysis would significantly improve the paper. I strongly encourage the authors to resubmit the paper once this has been addressed.